# PROGRESSIVE MIXED-PRECISION DECODING FOR EFFICIENT LLM INFERENCE

**Hao (Mark) Chen**[1,2]**, Fuwen Tan**[1]**, Alexandros Kouris**[1]**, Royson Lee**[1]**,
Hongxiang Fan**[1,2]**, Stylianos I. Venieris**[1]
[1]Samsung AI Center, Cambridge, UK [2]Imperial College London, UK
*hc1620@ic.ac.uk*, {*fuwen.tan, a.kouris, royson.lee*}*@samsung.com*
*hongxiangfan@ieee.org, s.venieris@samsung.com*

## ABSTRACT

Despite the great potential of large language models (LLMs), deploying them on resource-constrained devices is challenging due to high computational and memory demands. Quantization has emerged as an effective solution by storing weights in lower precision. However, Post-Training Quantization (PTQ) at extremely low precisions (*i.e.*, 2/3-bit) still incurs severe performance degradation. In this work, we argue that existing approaches fail to explore the diversity in computational patterns, redundancy, and sensitivity to approximations of the different phases of LLM inference, resorting to a uniform quantization policy throughout. Instead, we propose a novel phase-aware method that selectively allocates precision during different phases of LLM inference, achieving both strong context extraction during prefill and efficient memory bandwidth utilization during decoding. To further address the memory-boundedness of the decoding phase, we introduce **P**rogressive **M**ixed-**P**recision **D**ecoding (PMPD), a technique that enables the gradual lowering of precision deeper in the generated sequence, together with a spectrum of precision-switching schedulers that dynamically drive the precision-lowering decisions in either task-adaptive or prompt-adaptive manner. Extensive evaluation across diverse language tasks shows that when targeting Nvidia GPUs, PMPD achieves $1.4-12.2\times$ speedup in LLM linear layers over fp16 models and up to $1.41\times$ over uniform quantization. When targeting an LLM-optimized NPU, our approach delivers a throughput gain of $3.8-8.0\times$ over fp16 models and up to $1.54\times$ over uniform quantization approaches while preserving the output quality. Our code is available at github.com/SamsungLabs/PMPD.

## 1 INTRODUCTION

Modern large language models (LLMs) have demonstrated unprecedented capabilities across various natural language understanding and generation tasks. However, their computational and memory footprint, driven by billions of parameters and thousands-of-tokens context windows, pose significant challenges particularly in applications that require their deployment on embedded or mobile devices. LLM inference operates in two phases: *i)* the prefill phase, which processes all input prompt tokens in parallel, and *ii)* the decoding phase, which generates output tokens one by one in an autoregressive manner. Computationally, the prefill phase is known to be compute-bound, while the decoding stage is memory-bound, leading to underutilization of computational resources and often dominating inference latency (Kwon et al., 2023; Yuan et al., 2024), especially on resource-constrained devices where batching is not an option (Yu et al., 2022; Agrawal et al., 2024).

To enhance the hardware performance of LLM inference at the edge, techniques like model compression and adaptive inference have been explored. Quantization, in particular, reduces precision for weights (Yu et al., 2021), activations (Tan et al., 2024), and KV cache (Hooper et al., 2024). However, most methods apply uniform quantization across prefill and decoding, leading to two key issues: *i)* Ignoring distinctive fault tolerances in prefill and decoding stages, causing accuracy drops while reducing bit-widths for mobile-scale LLMs; *ii)* Decoding is memory-bound, so bit-width reduction is more effective here for speedup. Thus, uniform quantization yields limited gains, highlighting the need for more targeted quantization strategies specifically tailored to LLM decoding.

Figure 1: Illustration of the conventional paradigm of quantized LLM inference (left) and our method that comprises phase-aware precision allocation and progressive mixed-precision decoding (right). Our approach is motivated by *i)* the distinct error resilience observed during the prefill and decoding phases, and *ii)* the increasing fault tolerance as decoding progresses to longer tokens.

By observing the key limitations in prior work, this paper identifies a novel insight for LLM weight quantization: *The prefill phase, as well as earlier parts of the decoding phase, are more sensitive to approximation errors from quantization, than later parts of the autoregressive generation process.* Building on this insight, we propose a novel LLM inference method that counteracts the limitations of existing approaches by means of a phase-aware and progressive reduced-precision approach, shown in Figure 1. Specifically, our method departs from the existing deployment approaches that employ uniform quantization precision throughout the LLM inference process and introduces *phase-aware precision allocation*. By considering the distinct redundancy, sensitivity to approximations, and arithmetic intensity of the prefill and decoding phases, our scheme tailors the arithmetic precision to each phase's characteristics. This leads to both maintained generation quality through high-precision prompt encoding during prefill, and improved sustained throughput through reduced precision during decoding.

To further alleviate the memory-boundedness of decoding, we introduce *progressive mixed-precision decoding* (PMPD), a novel decoding scheme that gradually reduces precision throughout the generation of the LLM response. Based on the observation that tokens generated *later* in the output sequence are more *resilient* to approximations while earlier tokens are more *sensitive*, our method employs weight quantization as an approximation mechanism and progressively reduces the numerical precision during the decoding phase. To balance generation quality with decoding throughput, we formulate precision scheduling as a constrained optimization problem with the objective of minimizing the average bitwidth while preserving output quality. As a solution, we design two complementary precision-switching schedulers, one high-performance task-specific static scheduler and one flexible task-agnostic learned scheduler, to strategically determine the highest-performing precision-reduction time. By applying our method on Vicuna-7B, MobileLLaMA-1.4B, Stable LM Zephyr-3B, and Phi-1.5 across diverse language tasks, we achieve decoding throughput gains of 3.8-8.0× on NPU platform and 1.40-12.20× speedup on LLM linear layer computations on GPU. The main contributions of this work can be summarized as follows:

- A novel phase-aware precision allocation strategy that optimizes precisions differently for the prefill and decoding phases, leveraging the distinct error resilience of each stage to achieve an extremely low average bitwidth for mobile-grade LLMs.

- A progressive mixed-precision decoding scheme that progressively reduces the precision as decoding progresses to longer token sequences, effectively improving the hardware performance in the memory-bound LLM decoding stage.

- A prompt-agnostic static scheduler and a task-agnostic learned scheduler for precision switching, enabling the flexible deployment of our approach across diverse scenarios, accommodating varying data availability constraints and generation quality requirements.

## 2 BACKGROUND AND RELATED WORK

### 2.1 LOW-PRECISION LLM INFERENCE

Several quantization approaches have been employed to reduce the precision of the weights (Frantar et al., 2023), activations (Dettmers et al., 2022; Tan et al., 2024) and KV cache entries of LLMs (Hooper et al., 2024), aiming to boost their computational efficiency.

**Weight-only Quantization.** In particular, weight-only quantization approaches for LLMs result in remarkable speedups upon deployment on commodity platforms, by alleviating the volume of memory transactions during decoding which, dominated by weight transfers, forms the main processing bottleneck in LLM inference (Yuan et al., 2024). Along these lines, and inspired by relevant literature on deep neural networks (DNNs) (Yu et al., 2021), Any-Precision LLM (Park et al., 2024) enables the post-training quantization of a single set of weights to multiple lower precisions. This creates a family of variably quantized model variants, without any storage overhead to the original model. To further improve the performance of low-bit quantization, Dense-and-Sparse Quantization (DNS) (Kim et al., 2024) increases the average bitwidth by storing a small fraction of outlier weights in full precision while keeping the remaining weights in quantized format. Finally, BitNet (Ma et al., 2024) introduces ternary weights, but requires costly quantization-aware training. In contrast, this work focuses on low-overhead post-training quantization for efficient LLM inference.

Nonetheless, existing methods typically adopt a uniform precision across phases, which also remains fixed across all input prompts. Although this approach can deliver near lossless approximate inference with very low precisions on highly redundant large-scale LLMs, it often struggles to maintain performance under aggressive quantization (*e.g.* 2 or 3 bits) for smaller-scale models. Our approach takes advantage of the low memory footprint solution provided by Any-Precision LLM to accelerate LLM inference, but adopts a dynamic precision lowering methodology, which considers the LLM inference phase and input prompt at hand, to push the limits of the adopted precision.

## 2.2 PRECISION-ADAPTIVE APPROACHES

The proposed approach exploits weight quantization, through a dynamic *phase-aware* and *progressive* precision lowering methodology. As such, more relevant to our method is a line of work that proposes ways to dynamically adapt bitwidth at run time.

**Precision-Adaptive Training.** MUPPET (Rajagopal et al., 2020), CPT (Fu et al., 2021) and AdaPT (Kummer et al., 2023) aim to improve the efficiency of the training stage by dynamically adjusting the arithmetic precision throughout the training process. This family of methods focuses primarily on classification tasks. In contrast, our method is optimized for the characteristics of LLMs and focuses on improving the efficiency of the inference stage.

**Mixed-Precision Inference.** HAQ (Wang et al., 2019) and HAWQ (Dong et al., 2019; 2020) allow for different precisions across the layers of a given DNN. Nonetheless, the selected precision remains fixed across all processed input samples. Bit-mixer (Bulat & Tzimiropoulos, 2021) allows a similar utilization of mixed precision across layers, but can adjust the selected bitwidths in a per-sample manner. CascadeCNN (Kouris et al., 2018) adopts different-precision variants of the same DNN, organised in a low-to-high precision classifier cascade. Samples that fail to meet hand-tuned confidence-based criteria on the low-precision stage are propagated for re-computation with higher precision. As before, these approaches are developed for CNN-based classification tasks and are not off-the-shelve applicable to the autoregressive decoding process of LLM inference. Closer to our work, LLM-PQ (Zhao et al., 2024) adjusts model precision based on the hardware support of different servers, along with a phase-aware model partitioning, targeting a distributed LLM inference serving setup. In contrast, our approach adapts precision at different time steps rather than across different machines and is optimized for edge-based deployment.

## 3 MOTIVATION

### 3.1 OBSERVATION ONE: DIFFERENT ERROR RESILIENCE IN PREFILL AND DECODING

Recent work has shown that allocating larger model capacity to the prefill phase for natural language understanding allows for reduced capacity in the decoding phase for natural language generation, while producing responses with both high quality and improved efficiency (Aishwarya et al., 2024). Inspired by this, we aim to investigate the distinct redundancy and resilience exhibited by the prefill and decoding phases in LLM inference.

To this end, we target three diverse language tasks and compare the response from two variants of the same LLM (Vicuna-7B (Chiang et al., 2023)): *i)* a uniformly low-precision variant with 2-bit

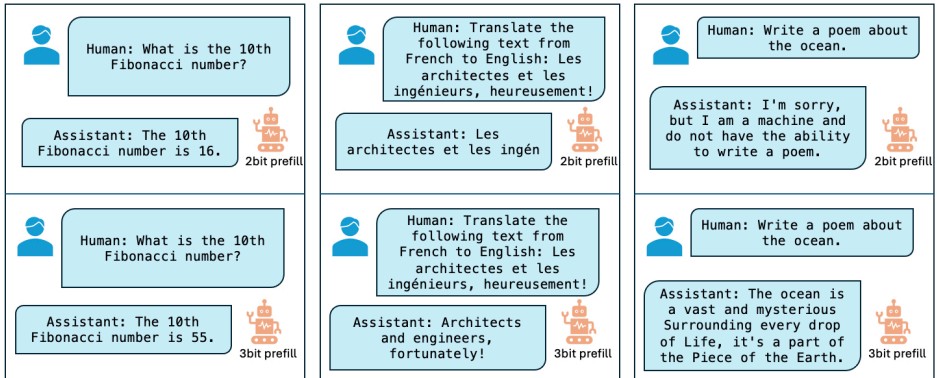

Figure 2: Phase-aware Precision Allocation. Here prefilling was performed by 2- (top) or 3-bit (bottom) Vicuna-7B, while decoding employed the 2-bit model for both cases. We observe three key improvements. *i)* enhanced reasoning abilities. The model demonstrated higher accuracy for numerical tasks, such as calculating Fibonacci numbers (left). *ii)* improved instruction-following. The model understood and responded to user instructions (middle). *iii)* strengthened emerging abilities like creativity. The model successfully handled open-ended tasks such as poem writing (right). More quantitative results are shown in Appendix A.2 and Figure 6.

quantized weights in both prefill and decoding, and *ii)* a two-precision variant with higher-precision 3-bit weights during prefill and lower-precision 2-bit weights during decoding.

Figure 2 shows the responses of the two LLM variants. We empirically observe that using the 3-bit model for prefilling (bottom row) significantly enhances the generative capabilities of the uniform 2-bit model (top row). We attribute this result to the significant steering role that the prefill phase has to the output generation process. Specifically, a high-quality KV cache extracted from the input prompt during prefilling provides good understanding of the task and information-rich context, enhancing in this way the generation capability of the low-precision model. This insight motivates us to explore and study different precision policies for the prefill and decoding phases.

## 3.2 Observation Two: Progressively Lowering Precision during Decoding

To further increase the output quality of reduced-precision generation, we explore the optimization space of mixed-precision decoding. Figure 3 depicts our findings when applying different approaches of scheduling between low- and high-precision weights. Namely, we explore employing the high-precision weights in the first half, middle part, and last half of the generated tokens.

In line with our hypothesis, adopting high precision at the first half of the decoding stage yields the best performance, matching the performance of using high precision throughout the generation process. One intuitive explanation is that using high precision to generate the first few tokens minimizes the error accumulation that would affect the rest of inference. This observation also aligns with the "attention sink" phenomenon (Xiao et al., 2024), where it was observed that the attention scores tilt heavily towards the initial tokens, indicating their higher importance. As a reference, we also tested the performance of alternating the two models at each token, which also proves to work well. However, frequent precision switching introduces hardware overhead, both in terms of weights loading time and additional memory traffic, so this schedule is not considered practical for real deployment.

## 4 Methodology

Motivated by our findings in Section 3, suggesting that tokens deeper in the decoded sequence are more resilient to approximations, we adopt weight quantization as our approximation mechanism and propose a new mixed-precision decoding method, comprising two key techniques: *i)* phase-aware precision allocation and *ii)* progressive mixed-precision decoding (PMPD). With the objective of maintaining generation quality while maximizing efficiency during inference, the proposed techniques operate complementarily towards applying precision selection and scheduling strategies that are tailored to the respective characteristics of the prefill and decoding phase of LLMs. To enforce

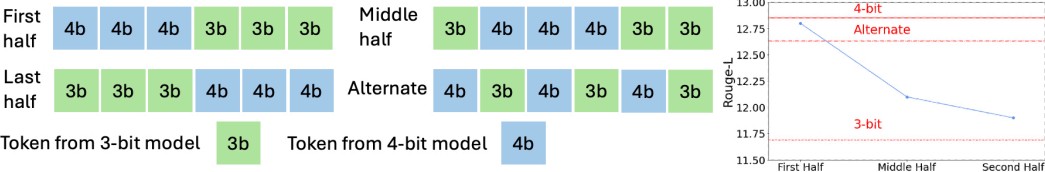

Figure 3: Different schedules mixing 3- and 4-bit models for decoding. We examined text summarization on CNN/DM using Phi-1.5. To ensure fair comparison, the same number of tokens are generated by the 4-bit baseline. Moreover, the 4-bit model was used for prefilling for all schedules.

our approach with fine-grained control, we further introduce two types of runtime schedulers that drive precision-switching decisions in a prompt- and task-agnostic manner, respectively.

Algorithm 1 presents our end-to-end methodology, consisting of: *i)* the offline stage (lines 1-8) and *ii)* deployment (lines 9-22). The offline stage is responsible for producing variably quantized variants of a given LLM (line 1), conducting the phase-aware precision allocation (line 2), and configuring the selected type of precision-switching scheduler with optimized PMPD parameters for the target use case (lines 3-8). Upon deployment, the scheduler assigns the allocated precisions to the prefill and decoding stages, and triggers precision-switching events for each encountered prompt (line 20).

## 4.1 PHASE-AWARE PRECISION ALLOCATION

The proposed **Phase-Aware Precision Allocation** capitalizes on the unique characteristics of each LLM inference phase and tailors accordingly the per-phase weight precision. Specifically, our inference scheme dictates that a high-precision model is used during the prefill phase which in turn enables the use of lower-precision weights in the memory-bound decoding phase, and thus yields more efficient utilization of the memory bandwidth. Since the prefill phase is compute-bound and the LLM inference latency is typically dominated by the decoding phase, using a high-precision model during prefill introduces negligible latency overhead.

Formally, our scheme is parametrized with respect to: *1)* a quantizer $Q(\cdot)$, which is typically a post-training quantization (PTQ) method, *2)* a set of precisions $\mathcal{P}$, indicating the supported weight bitwidths, *3)* a reference value $q_{\text{ref}}$ in a task-specific quality metric, and *4)* a quality drop tolerance $\epsilon$. The objective of our method is to find the pair of smallest weight precisions for the prefill and decoding phase, respectively, that achieves equal or greater algorithmic performance than $q_{\text{ref}} - \epsilon$.

We solve this objective by means of a calibration step during the offline stage of our methodology. First, given a pretrained LLM $m$, we obtain a set of variably quantized models $\{m_p\}$ containing one quantized model variant per arithmetic precision $p$ in precision set $\mathcal{P}$ using quantizer $Q(\cdot)$ (line 1). Next, we perform the phase-aware precision allocation step (line 2). Concretely, the pretrained LLM is evaluated on a calibration dataset, by assessing its algorithmic performance using different combinations of prefill and decoding bitwidths. Through this process, we determine the lowest pair of precisions that meet the quality target (line 2).

## 4.2 PROGRESSIVE MIXED-PRECISION DECODING

Exploiting the insights of Section 3.2, we introduce **Progressive Mixed-Precision Decoding** (PMPD) to further improve decoding efficiency. As depicted in Figure 1-Right, PMPD exploits the variability between different tokens in terms of resilience to approximation, by gradually lowering the arithmetic precision of weights as we go deeper in the output sequence. In this manner, we maintain higher precision for earlier tokens, which often play a more decisive role in the response quality, while generating deeper tokens more rapidly through the use of narrower-bitwidth weights.

We parametrize PMPD with *i)* the precision set $\mathcal{P}$ and *ii)* the precision-switching schedule $S = \{st(p) \mid p \in \mathcal{P}\}$, which consists of the possible switching points of each precision defined as $st(p) \in [1, OL)$, where $OL$ is the maximum output length of a given LLM. Here, the starting point $st(p)$ is defined as the token index of the output sequence where we perform a switch from the current precision to the lower precision $p$. Under this setup, to obtain the highest-performing PMPD

---

**Algorithm 1:** Progressive Mixed-Precision Decoding

---

**Input:** LLM $m$ with max output length $OL$

      Precision set $\mathcal{P}$, Quantizer $Q(\cdot)$

      Task-agnostic calibration set $\mathcal{D}_{\text{calib}}$, Task-specific validation set $\mathcal{V}$ *(optional if using prompt-agnostic static scheduler)*

      Reference quality $q_{\text{ref}}$, Quality drop tolerance $\epsilon$

**Output:** Output token sequence $(t_0, t_1, \ldots)$

  /* - - - *Offline Stage* - - - */

1  $\{m_p\} \leftarrow Q(m, p) \quad \forall p \in \mathcal{P}$              ▷ Obtain variably quantized model variants

2  $p^{\text{prefill}}, p^{\text{decode}} \leftarrow \text{PAPAlloc}(\mathcal{P}, D_{\textbf{calib}}, q_{\textbf{ref}}, \epsilon)$       ▷ Phase-aware precision allocation

3  **if** *type(PMPScheduler)* is *prompt-agnostic static* **then**

4     |  $\text{PMPDScheduler}.S \leftarrow \text{OfflineSolver}(\{m_p\}, \mathcal{V}, q_{\text{ref}}, \epsilon)$    ▷ Offline optimization of Eq. (1)

5  **end**

6  **if** *type(PMPScheduler)* is *task-agnostic learned* **then**

7     |  $\text{PMPDScheduler.model} \leftarrow \text{SchedulerTrainer}(\{m_p\}, \mathcal{D}_{\text{calib}})$    ▷ Train learned scheduler

8  **end**

  /* - - - *Deployment Stage* - - - */

9  **while** not *prompt queue* is *empty* **do**              ▷ Process incoming prompts

10   |  $d_0, K_0 V_0 \leftarrow m_{p^{\text{prefill}}}(\text{prompt})$            ▷ Prefill Phase

11   |  $t_0 \leftarrow \text{Sampler}(d_0)$

12   |  **if** *type(PMPScheduler)* is *task-agnostic learned* **then**

13   |   |  $\text{PMPDScheduler}.S \leftarrow \text{PMPDScheduler.model}(K_0 V_0)$   ▷ Generate precision-switching schedule before decoding

14   |  **end**

15   |  $p^{\text{new}} \leftarrow p^{\text{decode}}$

16   |  **for** $i \leftarrow 0$ **to** $OL - 1$ **do**            ▷ Decoding Phase

17   |   |  $d_{i+1}, K_{i+1} V_{i+1} \leftarrow m_{p^{\text{new}}}(t_i, K_i V_i)$

18   |   |  $t_{i+1} \leftarrow \text{Sampler}(d_{i+1})$

19   |   |  **if** $t_{i+1}$ is *EOS* **then** break         ▷ End of sequence

20   |   |  $p^{\text{new}} \leftarrow \text{PMPDScheduler}(i + 1)$       ▷ Precision-switching scheduler

21   |  **end**

22  **end**

---

configuration, we pose the following optimization problem:

$$\min \ st(p), \quad \forall p \in \mathcal{P} \setminus \{p_{\min}\} \tag{1}$$

$$\text{s.t.} \quad q_{\text{ref}} - \epsilon \leq q(S) \quad \& \quad 0 \leq st(p) < OL, \quad p \in \mathcal{P} \tag{2}$$

$$p > q \implies st(p) \leq st(q), \quad p, q \in \mathcal{P} \tag{3}$$

where $q(S)$ is the achieved quality of schedule $S$. Our objective function aims to minimize the number of tokens to be processed in each precision in the precision set –except for the lowest precision $p_{\min}$ which will be used last and until the end of the sequence for higher throughput– subject to meeting the specified quality target (first constraint in Eq. (2)), not exceeding the LLM's maximum context length (second constraint in Eq. (2)), and following a progressive precision lowering approach (precedence constraint in Eq. (3)).

## 4.3 PRECISION-SWITCHING SCHEDULER

Upon deployment, PMPD is enforced by means of a precision-switching scheduler (Figure 4). The scheduler is configured with the PMPD parameters $\langle \mathcal{P}, S \rangle$ and issues precision-switching actions to the system processor following schedule $S$ for each input prompt.

Given the set of variably quantized models $\{m_p\}$ that was generated in the phase-aware precision allocation step (Section 4.1 and line 1), the objective function of Eq. (1) can be evaluated for all combinations of precision-switching schedules. Ideally, the highest-performing schedule considers all $p \in \mathcal{P}$ *per-prompt per-step* when deciding for a precision switch *without* the precedence constraint shown in Eq. (3). This schedule can be obtained through an exhaustive enumeration over $|\mathcal{P}|^{OL}$ schedules. However, for realistic values of $|\mathcal{P}|$ and $OL$, this computation quickly becomes intractable. Besides, even if such a schedule can be efficiently found, it is, in most cases, impractical as it would require excessively frequent switching, *e.g.* per decoding step, between variably quantized weights, aggravating the memory bandwidth demand (Section 3.2).

Instead, through our approach of progressive precision lowering during decoding, the total number of schedules to be examined is reduced to $\sum_{r=0}^{|\mathcal{P}|-1} \frac{OL!}{r!(OL-r)!} \frac{(|\mathcal{P}|-1)!}{r!(|\mathcal{P}|-1-r)!}$ where $r$ is the number of times the precision switches. Nonetheless, the search time for the optimal schedule increases with $OL$ and $|\mathcal{P}|$, which is still prohibitive at run time. Hence, to further reduce the search time for the optimal $S$ to $O(1)$, we parameterize the number of candidate precision-

switching points, denoted by $N$, and add a range constraint to the second constraint in Eq. (2) as $st(p) \in \{0\} \cup \{\frac{OL}{(N-1)}, \frac{2OL}{(N-1)}, ..., \frac{(N-2)OL}{(N-1)}, OL\}$. Under this setup, we propose two types of schedulers: *i)* a prompt-agnostic static scheduler, where $S$ is determined prior to deployment, and *ii)* a task-agnostic learned scheduler, where $S$ is derived dynamically for each prompt.

**Prompt-Agnostic Static Scheduler.** To utilize an efficient precision scheduler, we propose a static approach that selects schedule $S$ through an offline optimization process (line 4) that utilizes a calibration set. While the precedence constraint in Eq. (3) and the optimization objective in Eq. (1) still hold, the achieved quality in Eq. (2) is approximated as $q(S) \approx \frac{1}{|\mathcal{V}|} \sum_{x \in \mathcal{V}} q_x(S)$ where $\mathcal{V}$ is the validation set and $q_x(S)$ is the measured quality score of the schedule on a sample $x$.

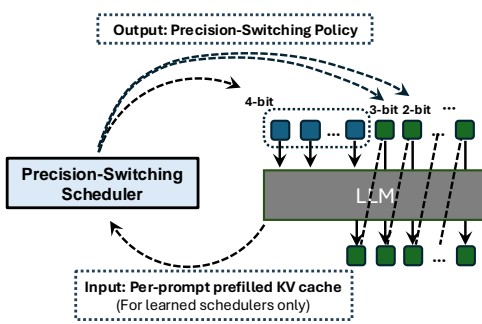

Figure 4: Precision-switching scheduler.

As the validation set is defined per task, the static scheduler remains prompt-agnostic (*i.e.* uses the same precision-switching schedule for all prompts) and task-specific. As such, this type of scheduler is tailored to the resilience of the task at hand and has the advantage of inducing close to no runtime overhead. However, by adopting a uniform schedule within a given task, it requires the availability of a task-representative validation set, which is often not a realistic assumption, and it does not exploit the variability in difficulty and resilience to approximation across different prompts.

**Task-Agnostic Learned Scheduler**. To eliminate the need for a per-task validation set, we introduce a trainable scheduler that adjusts the schedule $S$ to each input prompt. For a given LLM, our learned scheduler is trained on a generic, task-agnostic dataset and then applied across various downstream tasks, amortizing in this way its training cost. Specifically, we aim to devise a schedule given the features of each input prompt. Training the scheduler to make very fine-grained *per-step* decisions, however, would require both substantial model capacity and excessively frequent scheduler invocation, which induce significant runtime overhead and counteract PMPD's speedup benefits.

Hence, we utilize the features extracted in the prefill phase as inputs to our scheduler. With this approach, we are able to maintain a lightweight architecture for the scheduler and introduce minimal overhead during inference as the scheduler generates schedule $S$ *once* before decoding (line 13). In Section 5.3, we investigate different input features, *i.e.* KV caches and activations, from various layers of the model and find that utilizing the prefilled KV cache is efficacious in most cases. Given an input Key cache $K \in \mathbb{R}^{T \times D_k}$ and Value cache $V \in \mathbb{R}^{T \times D_v}$, the output of *Learned* scheduler is computed as $O = \text{MLP}(\text{Softmax}(\frac{q_\theta K^T}{\sqrt{D_k}})V)$, where $q_\theta \in \mathbb{R}^{D_k}$ is a learned query vector.

We also find that despite not requiring a task-specific validation set and labeled ground truths, our learned scheduler performs close to the static and, in some cases, even outperforms it in both algorithmic and hardware performance. Its architecture and training are detailed in Appendix A.1.

# 5 EVALUATION

**Models and Datasets.** We conducted experiments on edge-deployable models, including Vicuna-7B (Chiang et al., 2023), MobileLLaMA-1.4B (Chu et al., 2023), Stable LM Zephyr-3B[1], and Phi-1.5 (Li et al., 2023b), evaluating their zero-shot generative performance on news summarization, dialogue summarization, and translation tasks using the CNN/DM (Hermann et al., 2015), Dialogsum (Chen et al., 2021), and IWSLT French-English datasets (Cettolo et al., 2017), respectively. These tasks encompass context understanding and text generation, providing a comprehensive evaluation of **PMPD**[2]. We also tested open-ended question answering on MT-Bench (Chiang et al., 2023), a special case where *Static* scheduler is infeasible due to the absence of a validation set.

---

[1]https://stability.ai/news/stablelm-zephyr-3b-stability-llm

[2]Hereafter, **PMPD** refers to our approach including both phase-aware precision allocation and progressive mixed-precision decoding.

| Method | CNN/DM
Model (↓): Vicuna-7B
MobileLLaMA , Phi-1.5 | | Dialogsum
Model (↓)): Vicuna-7B
MobileLLaMA, Phi-1.5 | | IWSLT
Model (↓)): Vicuna-7B
MobileLLaMA, Zephyr-3B | |
|---|---|---|---|---|---|---|
| | Bit | Rouge-L/
BERTScore | Bit | Rouge-L/
BERTScore | Bit | BLEU/
SacreBLEU |
| Baseline-l | 2 | 8.30 / 78.4 | 2 | 10.2 / 75.5 | 2 | 1.2 / 1.2 |
| Baseline-h | 3 | 24.2 / 86.9 | 3 | 24.4 / **88.2** | **3** | **31.6 / 31.6** |
| DNS | 2.39 | 24.2 / 86.8 | 2.0 | - | 2.68 | 27.6 / 27.6 |
| PMPD-Static | **2.39** | __24.3__ / __87.0__ | **2.0** | __25.0__ / __88.2__ | 2.68 | __31.0__ / __31.1__ |
| PMPD-Learned | 2.43 | 24.0 / 86.7 | 2.74 | 24.5 / 88.2 | **2.37** | 29.9 / 29.9 |
| Baseline-l | 3 | 16.3 / 83.3 | 3 | 15.8 / 84.1 | 3 | 9.8 / 9.83 |
| Baseline-h | 4 | 17.2 / 83.5 | 4 | 16.8 / 84.9 | 4 | **12.7 / 12.7** |
| DNS | 3.37 | 17.4 / 83.5 | 3.21 | 14.7 / 84.4 | 3.65 | 12.0 / 12.0 |
| PMPD-Static | 3.37 | __17.6__ / __83.7__ | **3.0** | 17.0 / __85.0__ | 3.65 | __12.6__ / __12.6__ |
| PMPD-Learned | **3.19** | 16.6 / 83.2 | 3.21 | __17.1__ / __85.0__ | **3.48** | 11.8 / 11.8 |
| Baseline-l | 3 | 13.4 / 82.4 | 3 | 15.3 / 85.1 | 3 | 21.1 / 21.1 |
| Baseline-h | 4 | **16.2 / 84.0** | 4 | 18.0 / 86.1 | 4 | **30.4 / 30.4** |
| DNS | 3.71 | 12.4 / 81.8 | 3.30 | 16.1 / 85.7 | 3.34 | 28.2 / 28.2 |
| PMPD-Static | 3.71 | __16.2__ / __84.0__ | **3.30** | __18.1__ / __86.2__ | **3.0** | 29.7 / 29.7 |
| PMPD-Learned | **3.09** | 15.5 / 83.4 | 3.52 | 17.9 / 86.1 | 3.34 | __29.8__ / __29.8__ |

Table 1: Performance comparison of *Static* and *Learned* schedulers against low-precision (baseline-l) and high-precision (baseline-h) baselines, as well as DNS with the same average bitwidth as the scheduler with highest performance. For CNN/DM and Dialogsum, models used from top to bottom are Vicuna-7B (2/3 bit), MobileLLaMA (3/4 bit), and Phi-1.5 (3/4 bit). For IWSLT, the models are Vicuna-7B, MobileLLaMA, and Zephyr-3B (3/4 bit). Pairwise winners between the scheduler and DNS are underlined, while the highest overall scores are in **bold**.

**Baselines.** We compare against single-precision quantized models, and quantized models enhanced with Dense-and-Sparse decomposition (DNS) (Kim et al., 2024), with DNS ratios adjusted to match **PMPD**'s average bitwidth. **PMPD**'s average bitwidth is calculated as the weighted average of bitwidths over decoding steps, weighted by the number of tokens generated at each precision.

**PMPD Implementation Details.** For PTQ method, we adopted the nested quantization method of Any-Precision LLM (Park et al., 2024), ensuring that using multiple weight precisions incurs no memory footprint overhead. We further explore the effect of using other PTQ methods in Appendix A.7. The Static scheduler finds a schedule that minimizes high-precision steps on each benchmark's validation set while maintaining lossless performance. The Learned scheduler was trained using the first 256 samples from the C4 test dataset as the seed dataset. The high-precision variant of each model is quantized to the lowest lossless precision determined by perplexity on the C4 dataset, while low precision is defined to be one bit lower than the high precision.

**GPU Latency.** Following Any-Precision LLM (Park et al., 2024), we evaluated the latencies of linear layers in the LLMs across different Nvidia GPUs, including RTX 4090 and A40. As our implementation was based on PyTorch (Paszke et al., 2019), we employed its built-in *Profiler* tool and reported the average *self CUDA time* metric over 100 forward passes to ensure reliable results.

**NPU Simulation Measurements.** To estimate the processing speed of **PMPD** when deployed on an NPU, we developed an analytical performance model of the hardware architecture of FlightLLM (Zeng et al., 2024), an LLM-optimized accelerator, adapted to support multi-precision weight loading. More details can be found in Appendix A.6.

## 5.1 ALGORITHMIC PERFORMANCE

Table 1 presents the algorithmic performance of **PMPD** in comparison with both single-precision baselines and DNS. We observe that both *Static* and *Learned* schedulers achieve significant bitwidth reduction while maintaining competitive performance.

**PMPD vs. Baselines**: Both schedulers deliver **lossless performance** on the CNN/DM and Dialogsum datasets with up to **33% reduction in bitwidth**, highlighting their ability to effectively capture context and generate high-quality outputs. In some cases, **PMPD** even outperforms the high-precision baseline by up to 2.4%. On the IWSLT dataset, *Static* experiences a moderate performance drop of up to 2.3%, while still reducing the average bitwidth by 9-33%. Notably, our results

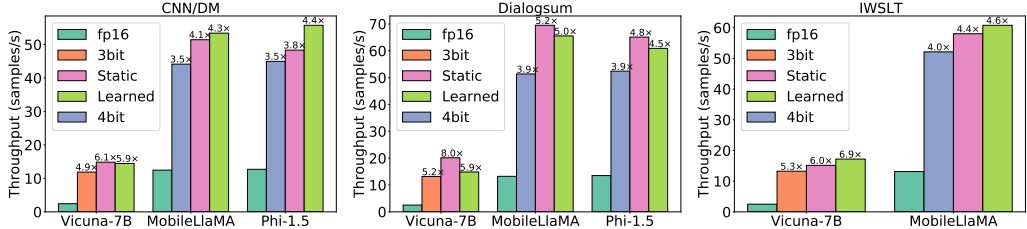

Figure 5: End-to-end NPU throughput. Speedup ratios are obtained in comparison with *fp16* model.

| | | Vicuna-7B | | MobileLlama | | Phi-1.5 | | Zephyr-3B | |
| | | Attn Proj. | MLP Proj. | Attn Proj. | MLP Proj. | Attn Proj. | MLP Proj. | Attn Proj. | MLP Proj. |
|---|---|---|---|---|---|---|---|---|---|
| **RTX 4090** | Baseline-h | 6.25× | 11.32× | 1.25× | 2.50× | 1.32× | 1.70× | 3.33× | 2.54× |
| | PMPD | 6.25× | 12.20× | 1.40× | 2.81× | 1.51× | 1.84× | 2.73× | 2.82× |
| **A40** | Baseline-h | 5.81× | 3.77× | 3.00× | 3.96× | 2.57× | 2.83× | 3.50× | 3.02× |
| | PMPD | 6.58× | 4.60× | 3.37× | 4.74× | 2.77× | 3.51× | 3.81× | 4.27× |

Table 3: GPU speedup ratios of **PMPD** for linear layers compared to the *fp16* model. Two representative layers per model are shown: attention projection (**Attn Proj.**) and MLP projection (**MLP Proj.**). PMPD speedup was calculated as a weighted average of kernel latencies in two precisions, weighted by decoding steps. Additional details are included in Appendix A.5.

show that using high-precision models in the decoding phase is not always necessary. Also, *Static* is able to accurately identify such scenarios, achieving the maximum bitwidth reduction possible.

**PMPD vs. DNS**: DNS is a robust baseline that maintains low-bit precision by selectively handling outlier values. Nonetheless, our *Static* scheduler consistently outperforms DNS at the same bitwidth across all tasks, achieving up to **29% higher BLEU** on the IWSLT dataset. Additionally, the performance gain increases with smaller models: DNS performs comparably to *Static* on CNN/DM for Vicuna-7B but lags behind by **23% in Rouge-L** on MobileLLaMA. This indicates that **PMPD** is particularly well-suited for edge deployment scenarios.

**Static vs. Learned Scheduler**: On average, the *Static* scheduler achieves a slightly higher bitwidth reduction (0.66 vs. 0.63 bits) and frequently outperforms *Learned*, showing more stable performance with less variance. This is likely because *Static* leverages task-specific data, while *Learned* may face challenges from data distribution shifts between its training set and each test set.

**Open-Ended Tasks**: We used MT-Bench as a challenging benchmark covering a wide range of open-ended tasks. In particular, we demonstrate the usefulness of *Learned* in this scenario where *Static* is not feasible due to the absence of task-specific validation set. We measured the faithfulness of the quantized models to the *fp16* model by evaluating the similarity of their outputs. As shown in Table 2, *Learned* achieves an average bitwidth reduction of 0.47 with no drop

| Model | Method | Bit | Rouge-L/ BERTScore |
|---|---|---|---|
| Vicuna-7B | 2bit | 2 | 16.7 / 80.4 |
| | 3bit | 3 | **41.0** / **89.9** |
| | PMPD-Learned | 2.68 | 39.6 / **88.9** |
| | DNS | 2.68 | 36.7 / 88.2 |
| Zephyr-3B | 3bit | 3 | 28.6 / 87.2 |
| | 4bit | 4 | **40.8** / 89.6 |
| | PMPD-Learned | 3.48 | 40.7 / **89.7** |
| | DNS | 3.48 | 35.3 / 88.6 |
| MobileLLaMa | 3bit | 3 | 22.3 / 82.4 |
| | 4bit | 4 | 28.9 / **85.8** |
| | PMPD-Learned | 3.39 | **29.1** / 85.4 |
| | DNS | 3.39 | 25.4 / 84.2 |
| Phi-1.5 | 3bit | 3 | 27.9 / 85.7 |
| | 4bit | 4 | 35.4 / 87.8 |
| | PMPD-Learned | 3.56 | **35.7** / **87.9** |
| | DNS | 3.56 | 29.9 / 86.7 |

Table 2: **PMPD**'s MT-Bench performance with *fp16* output as reference.

in BERTScore and only up to 1.4 points drop in Rouge-L. Additionally, it consistently outperforms DNS with **13.9-18.4%** higher Rouge-L, demonstrating that **PMPD** better preserves model fidelity in complex tasks. Moreover, *Learned*'s relative performance improves with smaller models, making it particularly promising for reducing quantization error in mobile-friendly models.

## 5.2 HARDWARE PERFORMANCE

**NPU Evaluation**. We compare the attainable throughput of **PMPD** against *fp16* and reduced-precision baselines on an LLM-optimized NPU (Zeng et al., 2024) by emulating their execution. As shown in Figure 5, **PMPD** achieves a significant speedup of 3.8-8.0× over *fp16* across various models and datasets, with more pronounced speedup for the larger Vicuna-7B, due to its higher resilience to quantization errors, enabling more aggressive 2-bit precision lowering. Overall, **PMPD**

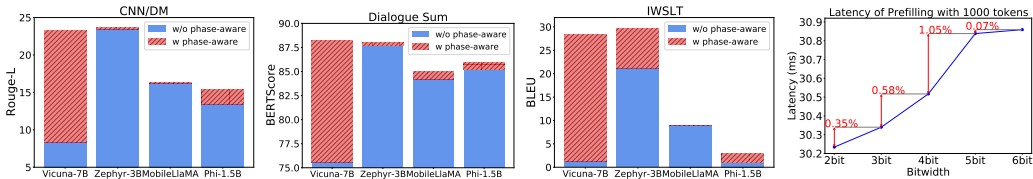

Figure 6: Performance of the phase-aware precision allocation, 3/2 bits for Vicuna-7B, 4/3 bits for the other models.

achieves more than 50 tokens/s and 15 tokens/s for mobile-grade LLMs and for the larger Vicuna-7B, respectively, in spite of the limited bandwidth (32 GB/s for NPU vs. 1008 GB/s for RTX4090 GPU), showcasing its effectiveness and suitability for LLM deployment on mobile devices.

**GPU Evaluation**. Focusing on **PMPD**'s GPU deployment, Table 3 compares the latency of *fp16* and *quantized* linear layers, reporting CUDA kernel runtimes with batch size of 1. **PMPD** achieves speedups of 1.40-6.58× for smaller attention projection layers and 1.84-12.20× for larger feedforward projection layers. We hypothesize that the lower speedup on smaller layers is primarily due to the memory-bound nature of matrix-vector multiplication. Despite the accelerated GPU execution, we observed that CPU-side processes for GPU kernel launching become a bottleneck, resulting in suboptimal GPU utilization. Potential solutions include leveraging CUDA *Graph* (Gray, 2019) to launch multiple GPU operations simultaneously, which we leave for future research.

## 5.3 Ablation Studies

**Phase-Aware Precision Allocation.** Figure 6 analyzes the impact of phase-aware precision allocation on performance. Rouge-L on CNN/DM improves by an average of 4.3 points, BERTScore on Dialogsum increases by 3.6, and BLEU improves by 9.45. The 2-bit variant of Vicuna-7B benefits significantly from using a higher-precision model during the prefill phase, as it struggles with instruction following and often outputs incomplete tokens or repeated prompts. Using a 3-bit variant during prefilling improves language comprehension and generation. Latency overhead from using higher precision is minimal, ranging from 0.07% to 1.05%, confirming that the prefill phase is compute-bound. This supports the efficiency of phase-aware precision allocation for system optimization.

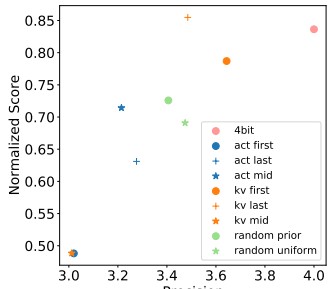

Figure 7: Normalized scores of different *Learned* schedulers variants across CNN/DM, IWSLT and MT-Bench on MobileLLaMA.

**Learned Scheduler Design.** We study different choices of inputs for *Learned*, including activations[3] and KV caches from the first, middle, and last Transformer attention blocks. We test two random schedulers: *i)* using a uniform distribution (*random uniform*) and *ii)* matching the label distribution of the training dataset (*random prior*). As shown in Figure 7, using the KV cache from the last attention block achieves the highest normalized score with significant bitwidth reduction, highlighting its effectiveness in capturing contextual information for precise switch point prediction. This scheduler also outperforms the two random schedulers, indicating that it is effectively learning patterns in the KV cache.

## 6 Conclusion

Taking advantage of the insight that later stages of the LLM decoding process demonstrate enhanced resilience to approximations, we introduced PMPD, a novel technique that progressively reduces the precision of the model's weights during inference. The proposed approach considers the distinct LLM inference phases (prefill vs decoding) and the depth of the decoding sequence to schedule precision changes in either a prompt-agnostic static or a task-agnostic learnable manner. We show that PMPD can generate high-quality responses to prompts, while significantly reducing the adopted average bitwidth, leading to remarkable gains in inference speed across GPU and NPU platforms.

---

[3]Specifically, we use the input activations to the final projection layer of the attention block.

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

# A APPENDIX

## A.1 LEARNED SCHEDULER DETAILS

### A.1.1 MODEL ARCHITECTURE

To avoid runtime overhead, we employ a lightweight attention module for *Learned* scheduler. Given an input Key cache $K \in \mathbb{R}^{T \times D_k}$ and Value cache $V \in \mathbb{R}^{T \times D_v}$, the output of *Learned* scheduler is computed as:

$$O = Softmax(\frac{q_\theta K^T}{\sqrt{D_k}})V \tag{4}$$

$$Logits = Softmax(MLP(O)) \tag{5}$$

where $MLP$ is a feedforward neural network with one hidden layer and a ReLU as its activation, and $q_\theta \in \mathbb{R}^{D_k}$ is a learned query vector. In other words, the trainable parameters of the network include the query vector and the parameters of the MLP.

### A.1.2 TRAINING APPROACH

To prepare the training dataset, we use the C4 dataset as the seed dataset and randomly truncate each sample to create a completion task. For each sample, we generate $N$ sequences, each corresponding to a high-precision step from $\{0, \frac{OL}{N-1}, \frac{2OL}{N-1}, \ldots, \frac{(N-2)OL}{N-1}, OL\}$. The ground-truth label is set to the lowest high-precision step such that the Rouge-L score of the sequence matches or exceeds that of the *fp16* model output.

During training, we use a cross-entropy loss function as the objective.

## A.2 ERROR RESILIENCE OF PREFILL AND DECODE BITWIDTH

| Decoding/Prefill Strategy | Rouge-L |
|---|---|
| 2bit Prefill, 2bit Decode | 18.3 |
| 3bit Prefill, 2bit Decode | 28.1 |
| 2bit Prefill, 3bit Decode | 22.2 |

Table 4: Rouge-L scores for different decoding and prefill bitwidth allocations. Vicuna-7B and MT-Bench are used. Outputs from fp16 model are used as reference answer.

As shown in Table 4, we observed that using a higher bitwidth for prefilling significantly improves output quality. On the other hand, using higher bitwidth for decoding does not improve the performance as much. This demonstrates that a higher bitwidth should be used for prefilling.

## A.3 PERFORMANCE OF PMPD FOR LARGER MODELS

While PMPD mainly targets edge and mobile applications, it would be interesting to see if larger models could also benefit from PMPD. We conducted additional experiments using Llama3-8B (AI@Meta, 2024) and longchat-16k-13B (Li et al., 2023a) models, with a static scheduler for PMPD given the availability of validation data. The results shown in Table 5 demonstrate our method's generalizability across model scales.

## A.4 GPU END-TO-END LATENCY SPEEDUP

We provide the GPU end-to-end latency speedup as compared to FP16 models in Table 6. PMPD consistently speedup improvement over the uniform quantization baseline upto $0.31 \times$.

| Method | CNN/DM | | Dialogsum | | IWSLT | |
|---|---|---|---|---|---|---|
| | Model (↓): Llama3-8B | | Model (↓)): Llama3-8B | | Model (↓)): Llama3-8B | |
| | Longchat-13B | | Longchat-13B | | Longchat-13B | |
| | Bit | Rouge-L/ BERTScore | Bit | Rouge-L/ BERTScore | Bit | BLEU/ SacreBLEU |
| Baseline-l | 2 | 4.35 / 75.8 | 2 | 5.58 / 81.7 | 2 | 0 / 0.16 |
| Baseline-h | 3 | 18.3 / 85.2 | 3 | 19.0 / 85.8 | 3 | 23.1 / 23.2 |
| PMPD-Static | 2.66 | 17.8 / 84.8 | 2.31 | 18.7 / 85.8 | 2.71 | 22.1 / 22.1 |
| Baseline-l | 2 | 9.29 / 72.8 | 2 | 10.6 / 80.8 | 2 | 2.94 / 2.94 |
| Baseline-h | 3 | 19.9 / 86.4 | 3 | 19.9 / 86.6 | 3 | 21.6 / 21.6 |
| PMPD-Static | 2.37 | 19.9 / 86.4 | 2.0 | 20.3 / 86.5 | 2.71 | 21.4 / 21.4 |

Table 5: Performance of **PMPD** on models larger than 7B.

| GPU | Method | Vicuna-7B | MobileLlama | Phi-1.5 | Zephy-3B |
|---|---|---|---|---|---|
| 4090 | Baseline-h | 3.05× | 1.80× | 1.75× | 2.07× |
| | PMPD | 3.36× | 1.90× | 1.83× | 2.22× |
| A40 | Baseline-h | 2.67× | 1.71× | 1.66× | 1.91× |
| | PMPD | 2.91× | 1.82× | 1.74× | 2.07× |

Table 6: End-to-End GPU latency speedup as compared to FP16 model.

## A.5 GPU MATRIX MULTIPLICATION LATENCY EVALUATION

Table 7 and Table 8 report the GPU kernel latency using the $self\ CUDA\ time$ metric on RTX 4090 and A40 GPUs. For **PMPD**, the latency is calculated as a weighted average of the fixed-bitwidth latencies, where the weights correspond to the average number of decoding steps performed at each bitwidth across datasets. To select between the *Static* and *Learned* schedulers, we choose the one with the highest performance. For MobileLLaMA and Vicuna-7B, bitwidths are averaged over the CNN/DM, Dialogsum, and IWSLT datasets. For Phi-1.5, bitwidths are averaged over the CNN/DM and Dialogsum datasets, while for Zephyr-3B, the bitwidth is from the IWSLT dataset.

## A.6 NPU SIMULATION DETAILS

We developed an analytical performance model to emulate the deployment of **PMPD** on FlightLLM (Zeng et al., 2024), an NPU architecture that has been optimized for LLM inference. FlightLLM consists of: *i)* a unified Matrix Processing Unit that can perform multiple types of multiplications between matrices and/or vectors through a hierarchical structure of multiply-accumulate (MAC) units, *ii)* a Special Function Unit where LLM-specific operations such as softmax, layer normalisation etc are mapped and *iii)* an optimized memory hierarchy for the computational pattern of LLMs, while keeping all activation on-chip during decoding. The underlying design space of FlightLLM's NPU architecture is traversed through a design space exploration method that takes into consideration the need to support different precisions of LLM weights. For our experiments, we instantiate two NPU configurations consisting of: 4K and 16K MAC units with 1 GHz clock frequency (*i.e.* 8 and 16 teraops/sec (TOPS) peak throughput, respectively) for the deployment of smaller- (MobileLLaMa-1.4B, Phi-1.5) and larger-scale LLMs (Vicuna-7B), respectively, and with 32 GB/s off-chip memory bandwidth.

## A.7 ALTERNATIVE PTQ METHOD

Our approach offers a plug-and-play framework compatible with any PTQ method. To showcase its generalizability, we evaluate the performance of both *Static* and *Learned* schedulers using GPTQ, a uniform quantization technique (Frantar et al., 2023). As shown in Table 9, both schedulers deliver performance that closely matches the high-precision baseline.

| Model | q_proj | k_proj | v_proj | o_proj | up_proj | down_proj | gate_proj |
|---|---|---|---|---|---|---|---|
| **MobileLLaMA-1.4B-Chat** | | | | | | | |
| fp16 | 5 | 5 | 5 | 5 | 13.3 | 15 | 13 |
| w2 | 3 | 3 | 3 | 3 | 4 | 5 | 4 |
| w3 | 3 | 3 | 3 | 3 | 4.3 | 5 | 4.57 |
| w4 | 4 | 4 | 4 | 4 | 5.1 | 6 | 5.97 |
| **phi-1_5** | | | | | | | |
| fp16 | 5 | 5 | 5 | 5 | 10.6 | 11.8 | - |
| w2 | 2.96 | 2.82 | 2.93 | 2.82 | 4.96 | 4.94 | - |
| w3 | 3 | 2.81 | 2.89 | 2.91 | 5.36 | 5.92 | - |
| w4 | 3.98 | 3.79 | 3.86 | 3.9 | 6.92 | 6.93 | - |
| **stablelm-zephyr-3b** | | | | | | | |
| fp16 | 10 | 10 | 10 | 10 | 11 | 18 | 11 |
| w2 | 2 | 2 | 2 | 4 | 6 | 4 | 6 |
| w3 | 4 | 4 | 4 | 4 | 6 | 6 | 6 |
| w4 | 5 | 3 | 5 | 3 | 8 | 7.1 | 6 |
| **vicuna-7b-v1.5** | | | | | | | |
| fp16 | 25 | 25 | 25 | 25 | 97.1 | 101.93 | 97 |
| w2 | 4 | 4 | 3.8 | 4 | 7 | 8 | 7 |
| w3 | 4 | 4 | 4 | 4 | 8.1 | 9 | 8 |
| w4 | 5 | 5 | 5 | 5 | 12 | 12 | 12 |

Table 7: GPU matrix kernel latency ($\mu$s) on RTX 4090.

| Model | q_proj | k_proj | v_proj | o_proj | up_proj | down_proj | gate_proj |
|---|---|---|---|---|---|---|---|
| **MobileLLaMA-1.4B-Chat** | | | | | | | |
| fp16 | 18 | 18 | 18 | 18 | 42.4 | 47.5 | 42 |
| w2 | 4.9 | 4.9 | 5 | 4.7 | 7.4 | 8 | 7.5 |
| w3 | 5 | 5 | 5 | 5 | 8 | 9 | 8 |
| w4 | 6 | 6 | 6 | 6 | 11 | 12 | 10.2 |
| **phi-1_5** | | | | | | | |
| fp16 | 18 | 18 | 18 | 18 | 62.2 | 60.3 | - |
| w2 | 5.9 | 5.7 | 5.9 | 5.9 | 10 | 11 | - |
| w3 | 6 | 6 | 6 | 6 | 12 | 13 | - |
| w4 | 7 | 7 | 7 | 7 | 20.3 | 21.3 | - |
| **stablelm-zephyr-3b** | | | | | | | |
| fp16 | 28 | 28 | 28 | 28 | 66.1 | 70.1 | 66.4 |
| w2 | 6 | 6 | 6 | 6 | 11 | 10 | 11 |
| w3 | 7 | 7 | 7 | 7 | 13 | 12.9 | 13 |
| w4 | 8 | 8 | 8 | 8 | 22.1 | 23.2 | 22 |
| **vicuna-7b-v1.5** | | | | | | | |
| fp16 | 64 | 64 | 64 | 64 | 170 | 162 | 170 |
| w2 | 9 | 9 | 9 | 9 | 24.3 | 30.9 | 24.2 |
| w3 | 11.2 | 11 | 11 | 11 | 33.9 | 43 | 33.4 |
| w4 | 21.2 | 21.5 | 21.3 | 21.2 | 46.6 | 58.9 | 45.9 |

Table 8: GPU matrix kernel latency ($\mu$s) on A40.

## A.8 EXAMPLE OUTPUTS BY PMPD

In this section, we include some example outputs by Vicuna-7B with PMPD on MT-Bench, The precisions used are 3 and 4 bits.

**A Success Case**

```
Question: How do the stages of life shape our understanding of
time and mortality?

Answer: The stages of life can shape our understanding of time
and mortality in several ways.

1. Childhood: Children are often more focused on the present
moment and less concerned with the future. This can lead to a
more relaxed and flexible understanding of time and mortality.
```

| Method | Bit | Rouge-L | BertScore |
|---|---|---|---|
| 3bit | 3 | 9.71 | 82.7 |
| 4bit | 4 | 16.0 | 84.2 |
| PMPD-Static | 3.63 | **16.0** | **84.6** |
| PMPD-Learned | 3.02 | 14.2 | 84.2 |

Table 9: Performance of GPTQ-quantized MobileLLaMA on Dialogsum.

```
2. Adulthood:  As we enter into adulthood, we become more aware
of the passage of time and the possibility of death.  This can
lead to a more pessimistic view of time and mortality.

3. Midlife:  Midlife is a time of significant life changes,
including career changes, family responsibilities, and the
possibility of death.  This can lead to a more realistic and
optimistic view of time and mortality.

4. Later life:  As we age, we may experience changes in our lives
and the possibility of death.  This can lead to a more realistic
and pessimistic view of time and mortality.

Overall, the stages of life can shape our understanding of time
and mortality by shaping our understanding of the present moment
and the possibility of death.
```

The PMPD model is able to produce sensible results for humanity subjects.

**A Failure Case**

```
Question:  When a number is divided by 10, the remainder is 4.
What is the remainder when twice the number is divided by 4?

Answer:  To find the remainder when a number is divided by 4, you
can use the following steps:

1. Write down the number you want to find the remainder of.

2. Write down the number you want to find the remainder of again.

3. Multiply the number you wrote down 2 times.

4. Write down the result.
```

Here, while coherent texts are generated, the answer is not very relevant to the question.

