# OpenReview forum: "Progressive Mixed-Precision Decoding for Efficient LLM Inference"
_ICLR.cc/2025/Conference — ICLR 2025 Poster_

### Official Review · Reviewer_FUCY · 2024-11-03

**Soundness:** 3
**Presentation:** 4
**Contribution:** 3
**Rating:** 6
**Confidence:** 4

**Summary:**

The paper proposes phase-aware precision/quantization recipe which allocates higher bit depth for the prefill phase and lower/adaptive bit depth for the decoding phase in LLMs. They motivation is based off of previous work & representative examples shown in the paper that the prefill phase (language understanding phase) is likely to benefit from lower quantization errors than the decoding phase.
Additionally, authors propose a well executed "adaptive" precision strategy in the decoding phase, where the precision is reduced after every few steps of the generation.
The experiments on three small-scale mobile LLMs are show the effectiveness of the approach in terms of quality and latency/throughput of the kernels as compared to the other methods which use fixed bit-depth through out the generation.

**Strengths:**

It is very critical to address the memory-boundedness of small LLMs operating on mobile phones. Quantization is the most prevalent method for model compression but suffers from quality-latency tradeoff. The paper proposes "adaptive" precision in the decoding stage, by training a "precision scheduler", which is very interesting. It exploits the fact that later half of the tokens in the generation process can afford higher quantization error in the weights. Evaluation shows that the proposed methods achieves better (or at-least maintains) better quality than high bit-depth quantization methods, while increasing the throughput which is critical for mobile performance of small LLMs. The motivation is clear and backed with relevant prior work.

**Weaknesses:**

1) There is a lot of repetition of concepts in the first few pages of the paper. The same idea is explained several times in the first three sections. It maybe better to show more representative examples similar to Figure 2 -- perhaps some high level stats/evals on how different tasks/models perform when changing the precision in the decode/prefill phase.
2) The work misses several critical LLM benchmarks around Math, reasoning and especially long context. It would be interesting to see the effect of the proposed approach on a wider range of LLM evals.
3) This experimentation section fails to provide more statistics on evaluation, for example, what is the average decode length for the three tasks for the thee models, what is the impact of changing N (the number of "switches" in precision during decoding phase). It is not entirely clear what precision the scheduler choses and for how long at each decoding step, -- for three different tasks and the three models.

**Questions:**

1) IMO the most interesting section of the paper is the scheduler design in Section 4.3. As such a lot of core ideas/implementation details of the scheduler are relegated to the Appendix. It maybe better to add scheduler model/training design to the main paper, while reducing redundant text in the introduction, methodology and the motivation section.
2) When adding the constraint in line 325-327, it's not entirely clear how is N chosen, is this a hyper-parameter, what is recommended value? Does it depend on task/model that were studied? A detail ablation can be performed in the experiment section, but a brief summary for practioners can be provided here.
3) Table 1 caption says the phi-1.5 models are 2/4 bit, but the table itself shows they are 2/3 bit?

---

> ### Author Response · Authors · 2024-11-21
>
> Dear Reviewer FUCY,
>
> We thank you for your time and your detailed feedback. Please check our reponses below.
>
> Q1.1: replacement of repetition of concepts with details of scheduler model/training design.
>
> Thank you for the feedback. We will revise the paper to make it concise and include scheduler model/training details in the main paper.
>
> Q1.2: high level stats/evals on how different tasks/models perform when changing the precision in the decode/prefill phase.
>
> We provide  experiments with high-precision  decoding and low-precision prefilling for Vicuna-7B on MT-Bench.Note that this analysis is also included in our response to Reviewer **RDtY**
>
> | Decoding/Prefill Strategy | Rouge-L |
> | :---- | :---- |
> | 2bit Prefill, 2bit Decode | 18.3 |
> | 3bit Prefill, 2bit Decode | 28.1 |
> | 2bit Prefill, 3bit Decode | 22.2 |
>
> In the paper, we also presented relevant quantitative results of more models and tasks in Figure 3, Table 1, and FIgure 6\. We will add include more discussions in the revised paper.
>
> Q2: LLM benchmarks around Math, reasoning and especially long context.
>
> We added gsm8k for Math/Reasoning and LongBench for long context evaluation. Note that for other models we tested, given their small size and limited context window length adopted by  edge-friendly models, they did not perform well on Math and long-context tasks even in full precision.
>
> | Model | Method | gsm8k  |  | LongBench (multi-news) |  |
> | :---- | :---- | :---- | :---- | :---- | :---- |
> |  |  | bit | accuracy | bit | Rouge-L/BERTScore |
> | longchat-13B-16k | 2bit | 2 | 3.1 | 2 | 0.942/46.5 |
> |  | 3bit | 3 | 7.1 | 3 | 17.6/84.6 |
> |  | PMPD | 2.9 | 6.9 | 2.7 | 17.4/84.0 |
>
> Q3: more statistics on evaluation like average decode length
>
> We provide the average decode length for three tasks and three models here. The stat is in format {precision: average decode length}. For example, {'3': 39.0, '2': 65.4} means 3-bit model is used to generate 39 tokens and 2-bit model is used to generate 65.4 tokens on the average.
>
> CNN/DM:
>
> | Model | high-precision | low-precision | Static scheduler | Learned scheduler |
> | :---- | :---- | :---- | :---- | :---- |
> | Vicuna | {'3': 111.5} | {'2': 77.6} | {'3': 39.0, '2': 65.4} | {'3': 46.3, '2': 58.3} |
> | MobileLlama | {'4': 245.9} | {'3': 220.6}  | {'4': 83.0, '3': 158.6} | {'3': 189.7, '4': 45.7} |
> | Phi | {'4': 236.9} | {'3': 252.9} | {'4': 163.2, '3': 74.9} | {'4': 21.2, '3': 228.6} |
>
> Dialoguesum:
>
> | Model | high-precision | low-precision | Static scheduler | Learned scheduler |
> | :---- | :---- | :---- | :---- | :---- |
> | Vicuna | {'3': 32.8} | {'2': 26.6} | {'2': 32.6} | {'3': 24.6, '2': 8.2} |
> | MobileLlama | {'4': 33.0} | {'3': 32.8} | {'3': 32.9}} | {'3': 26.0, '4': 6.9} |
> | Phi | {'4': 33.0} | {'3': 32.9} | {'4': 10.0, '3': 23.0} | {'4': 17.4, '3': 15.6} |
>
> IWSLT:
>
> | Model | high-precision | low-precision | Static scheduler | Learned scheduler |
> | :---- | :---- | :---- | :---- | :---- |
> | Vicuna | {'3': 57.3} | {'2': 40.6} | {'3': 38.8, '2': 18.4} | {'3': 20.7, '2': 36.3} |
> | MobileLlama | {'4': 60.0} | {'3': 56.6} | {'4': 39.0, '3': 20.9} | {'4': 29.0, '3': 30.4} |
> | Zephyr | {'4': 59.4} | {'3': 59.9} | {'3': 59.6} | {'4': 20.1, '3': 39.4} |

---

> ### Author Response · Authors · 2024-11-21
>
> Q4: effect of different N and suggestion for the choice of N.
>
> N is a user-defined hyperparameter. As discussed in Section 4.3, larger N leads to larger search space, while leading to potentially larger bitwidth reduction or higher performance. The optimal N depends on both task and model. We empirically found that N=4 is a reasonable choice that balances the performance and search complexity.
>
> We conducted an ablation study to evaluate the impact of N on the maximum bitwidth reduction while maintaining a performance drop of less than 2% as compared to high-precision model. The results show that N=4 achieves a good balance, enabling significant bitwidth reduction while meeting performance requirements for most tasks. Note that for Dialoguesum, solely using the low-precision model during decoding achieves lossless performance so the maximum bitwidth reduction is already achieved at N=2. For IWSLT, when N \<= 3, a precision switch step that allows for performance drop within 2% couldn’t be found, so the high-precision model is used throughout, leading to no drop in both bitwidth and performance.
>
> | Model | N | Dialoguesum |  | IWSLT |  |
> | :---- | :---- | :---- | :---- | :---- | :---- |
> |  |  | Bitwidth Reduction | Performance Drop (%) | Bitwidth Reduction | Performance Drop (%) |
> | Vicuna | 2 | 1.0 | 0.0 | 0.0 | 0.0 |
> |  | 3 | 1.0 | 0.0 | 0.0 | 0.0 |
> |  | 4 | 1.0 | 0.0 | 0.32 | 1.8 |
> |  | 5 | 1.0 | 0.0 | 0.32 | 1.8 |
> |  | 6 | 1.0 | 0.0 | 0.32 | 1.8 |
> | MobileLlama | 2 | 1.0 | 0.0 | 0.0 | 0.0 |
> |  | 3 | 1.0 | 0.0 | 0.0 | 0.0 |
> |  | 4 | 1.0 | 0.0 | 0.35 | 0.8 |
> |  | 5 | 1.0 | 0.0 | 0.35 | 0.8 |
> |  | 6 | 1.0 | 0.0 | 0.35 | 0.8 |
>
> Q5: Inconsistency of bitwidths specified in the table and caption for phi-1.5.
>
> Thank you for pointing this out. This is indeed a typo. It should be 3/4 bit in both the table and caption. We will correct it in the revised paper.

---

> > ### Comment · Area_Chair_eGvj · 2024-11-24
> > **discussion**
> >
> > Dear Reviewer FUCY -- could you please look at the authors' rebuttal and acknowledge that you've read it?  Also, if you have any further questions for the authors, please let them know ASAP.

---

> > > ### Comment · Reviewer_FUCY · 2024-11-26
> > > **Thanks for the rebuttal**
> > >
> > > Thanks for providing details and additional experiments. Most of the concerns and questions have been addressed, and I bumped my scores to reflect the same.

---

> > ### Comment · Reviewer_FUCY · 2024-11-26
> > **Thanks for addressing the questions and providing additional experiments**
> >
> > Thanks for providing details and additional experiments. Most of the concerns and questions have been addressed, and I bumped my scores to reflect the same.

---

### Official Review · Reviewer_TCS8 · 2024-11-03

**Soundness:** 3
**Presentation:** 3
**Contribution:** 3
**Rating:** 6
**Confidence:** 3

**Summary:**

Authors propose a novel phase-aware method that selectively allocates precision during different phases of LLM inference: use higher precision during prefill stage and then reduce precision for efficient memory bandwidth utilization during decoding.
They introduce Progressive Mixed-Precision Decoding (PMPD) which controls gradual lowering of weights precision in the generated sequence with task-adaptive or prompt-adaptive manner. It is weights only post training quantization(PTQ) which adaptively controls weights precision depending on decode stage (prefill, decode) and decoded token. They use  Any-Precision LLM (Park et al., 2024), as PTQ which keeps multiple weight precisions with no memory footprint overhead.

Authors show that PMPD achieves 1.4−12.2x speedup in LLM linear layers over fp16 models (on Nvidia GPUs).
Also throughput gain of 3.8−8.0x over fp16 models and up to 1.54× over uniform quantization (on NPU) approaches while preserving the output quality.

**Strengths:**

PMPD vs. Baselines: Both schedulers deliver lossless performance on the CNN/DM and Dialogsum datasets with up to 33% reduction in bitwidth, highlighting their ability to effectively capture context and generate high-quality outputs.

Novelty in applying different precision for model weights depending on input token during decode mode.

Conducted thorough experiments on LLM edge-deployable models.

Produced End-to-end NPU throughput and Speedup comparison vs fp16 model. Shows that PMPD achieves a significant speedup of 3.8-8.0× over fp16 across various models and datasets.

Good ablation studies showing that  indicating that PMPD-Learned is effectively learning patterns in the KV cache

**Weaknesses:**

Q: in Table 1, PMPD-Static has better accuracy score in 11 cases; PMPD-Learned has better accuracy score in 3 cases; PMPD-Static uses less bits 5 cases; PMPD-Learned uses less bits 4 cases. It looks like PMPD-Learned under-performing on both accuracy and bits usage in comparison to PMPD-Static. Please analyze why PMPD-Learned underperforms compared to PMPD-Static. I see that PMPD-Learned is preferable in case when validation data are missing, because PMPD-Static can not be trained on data which are missing validation data sets.

Q: Authors report 1.4−12.2x speedup in LLM linear layers over fp16 models for PMPD methods. Please also add speed up in comparison to 3bit and 4bit baseline models (you show it in Figure 5 and Table 3, but do not report in the abstract of the paper).

Q: One of the goal of this paper is to show that proposed quantization approach has no accuracy drop in comparison to the float baseline model. Float baseline is missing in Table 1, so it is hard to make a conclusion that proposed methods do not drop accuracy in comparison to float baseline. Please include the float baseline results in Table 1 or explain why you chose not to include them (for example in Figure 5 you show fp16 latency of evaluated models). Same comment for Table 2.

Q: It would be great to add in Table 1, other state of the art post training quantization methods, e.g. :
* Fast and Efficient 2-bit LLM Inference on GPU: 2/4/16-bit in a Weight Matrix with Asynchronous Dequantization
* QuIP: 2-Bit Quantization of Large Language Models With Guarantees or (QuIP#: Even Better LLM Quantization with
Hadamard Incoherence and Lattice Codebooks)
* GPTQ: Accurate Post-Training Quantization for Generative Pre-trained Transformers

Please explain why above methods were not included as baselines, and how your method compares conceptually to these approaches if direct comparison is not feasible.

**Questions:**

Q : What quantization method is used for Baseline-l and Baseline-h?

Q: Why GPTQ are not presented as baselines in Table 1?

Q: How would you explain that PMPD-Static has better performance in comparison to PMPD-Learned on Table 1?

Q: Please explain what precision range was used for training PMPD shown on Table 1, e.g. (2bits, 3bits, 4bits) for all experiments or you use (2bits 3bits) for one one set of experiments and (3bits, 4bits) for another set of experiments?

Q: For PMPD method you will need to keep pre-computed e.g. 2bits 3bits weights and then switch between different precision depending on input token, please explain how do you avoid overhead of keeping weights with different set of precision in memory in comparison to a baseline method which keeps weights with one set of precision e.g. 3bits only?

Q: On Table 1, experiment on data Dialogsum, how do you explain that Baseline-l with 2bits has accuracy 10.2 / 75.5 but PMPD-Static with the same amount of bits = 2.0bits has better accuracy 25.0 / 88.2? Please explain what set of bits PMPD-Static uses, e.g. if it was trained with (2bits 3bits) then it means that PMPD-Static used only 2bits during inference, so that final number of bits is 2.0 (reported in Table 1).

---

> ### Author Response · Authors · 2024-11-21
>
> Dear Reviewer TCS8,
>
> We thank you for your time and your detailed feedback. Please check our reponses below.
>
> Q1: Comparison between PMPD-Learned and PMPD-Static in Table 1.
>
> Thanks for this insightful question. The key distinction is that PMPD-Static requires in-domain validation data, while PMPD-Learned operates in a zero-shot manner. The performance gap between these approaches can be attributed to two factors:
> Domain Shifts:
> While we used the C4 dataset to train the PMPD-Learned scheduler for better cross-task generalization, significant distribution shifts were observed between this seed dataset and the downstream tasks we evaluated on.
> Consistent Performance Gap Across Samples for a Single Task:
> Within individual tasks, we observed that the performance gaps between models of different precisions remained consistent across samples. This consistency suggests that a static schedule per task can effectively approximate the optimal precision scheduling policy, making PMPD-Static particularly effective.
>
> Q2: Including speedup with respect to 3 and 4-bit baselines to the Abstract.
>
> Thank you for the suggestion. We will include the number in the Abstract in the revised paper.
>
> Q3: Is PMPD lossless? How does it compare to unquantized fp16 models?
>
> PMPD introduces a precision allocation scheme that is independent of the underlying post-training quantization (PTQ) method. Whether it is lossless depends on the PTQ method it uses rather than its precision allocation scheme. In the table below, we provide further comparisons to the fp16 models, demonstrating that PMPD generally preserve the fp16-level accuracy in most scenarios.
>
> |  |  | CNN/DM (Vicuna-7B, MobileLlaMA, Phi-1.5)(Rouge-L/BERTScore) | DSUM (Vicuna-7B, MobileLlaMA, Phi-1.5) (Rouge-L/BERTScore) | IWSLT (Vicuna-7B, MobileLlaMA, Zephyr) (BLEU/SacreBLEU) |
> | :---- | :---- | :---- | :---- | :---- |
> | Vicuna-7B | fp16 | 24.8/87.1 | 26.7/88.8 | 33.2/33.2 |
> |  | PMPD | 24.3 / 87.0 | 25.0 / 88.2  | 31.0 / 31.1 |
> | MobileLlaMA | fp16 | 18.4/84.2 | 17.8/85.5 | 14.3/14.3 |
> |  | PMPD | 17.6 / 83.7 | 17.1 / 85.0 | 12.6 / 12.6 |
> | Phi-1.5/Zephyr | fp16 | 16.7/84.3 | 18.3/87.4 | 30.3/30.3 |
> |  | PMPD | 16.2 / 84.0  | 18.1 / 86.2  | 29.8 / 29.8 |
>
> Regarding MT-Bench results (Table 2): as we used the fp16 model outputs as reference answers, the Rouge-L and BertScore metrics for the fp16 model could become unreasonably high, making it unsuitable as a baseline for this specific evaluation.
>
> We will include more fp16 comparisons in the revised paper.
>
> Q4: Comparisons to more baselines.
>
> Please refer to our answer to Reviewer A5Dp for more discussions about the comparisons to other baselines, e.g. GPTQ.
>
> To summarize, PMPD introduces a precision allocation scheme that is independent of the underlying post-training quantization (PTQ) method. In our experiments, we primarily used SqueezeLLM as our PTQ approach. Direct comparisons with other PTQ methods (such as GPTQ [4] and QuIP [7]) would primarily reflect differences in the underlying quantization approaches—a comparison already reported in the SqueezeLLM paper [2], which demonstrated SqueezeLLM's superior performance compared to GPTQ [4] and QuIP [7].
>
> We also tried to include comparisons to other baselines [6] suggested by the reviewer, but did not manage to find any public implementation.
>
> We will include more discussions about the baselines in the revised paper.
>
> [2] SqueezeLLM: Dense-and-Sparse Quantization. ICML 2024.
>
> [4] GPTQ: Accurate Post-Training Quantization for Generative Pre-trained Transformers
>
> [6] Fast and Efficient 2-bit LLM Inference on GPU: 2/4/16-bit in a Weight Matrix with Asynchronous Dequantization.
>
> [7] QuIP: 2-Bit Quantization of Large Language Models With Guarantees.

---

> ### Author Response · Authors · 2024-11-21
>
> Q5: Quantization method for Baseline-l and Baseline-h
>
> We used SqueezeLLM with Dense-And-Sparse decomposition (DNS) as the baseline. Since PMPD also utilized SqueezeLLM (without DNS), this ensures a fair comparison. In this setting, our bitwidth reduction is solely attributed to PMPD’s precision allocation policy, which effectively removes redundancy during the decoding process.
>
> Q6: Precision range used for PMPD shown on Table 1.
>
> For Vicuna-7B, we used 3bit and 2bit. For all other models (MobileLlama, Phi-1.5, Zephyr-3B), we used 3bit and 4bit. This is because larger models like Vicuna-7B have lower quantization error and its 3bit version is almost lossless as suggested in SqueezeLLM[2].
>
> Q7: Memory overhead of precision switching.
>
> Please refer to our answer to Reviewer A5Dp for more discussions about the implementation and efficiency of precision switching.
>
> In summary, as described in Sec. 2.1, PMPD leverages AnyPrecision LLM [3], which enables efficient storage of M different precision models (1 to M-bit) using only M bits, while preserving model accuracy. This approach ensures PMPD requires no additional memory overhead. Furthermore, precision switching incurs negligible latency since no extra weight loading is required.
>
> We adopted AnyPrecision LLM to store models of different precisions, as detailed in their [paper](https://www.arxiv.org/pdf/2402.10517). AnyPrecision LLM enables storing up to n models of different precisions using only n bits, without compromising the performance of any model. Hence, our method does not incur additional memory overhead. For instance, when using both 3-bit and 2-bit models, the memory usage remains equivalent to that of a single 3-bit model.
>
> Q8: PMPD-Static with the same bitdiwth outperforms Baseline-l with 2bits
>
> For that specific experiment in Table 1, PMPD-Static uses 3bit for prefilling and 2bit only for decoding. Hence, the average bitwidth for decoding is 2. As shown in Figure 6, using a higher precision for prefilling increases the performance without incurring noticeable latency overhead.
>
> [2] SqueezeLLM: Dense-and-Sparse Quantization. ICML 2024.
>
> [3] Any-Precision LLM: Low-Cost Deployment of Multiple, Different-Sized LLMs. ICML 2024

---

### Official Review · Reviewer_A5Dp · 2024-11-03

**Soundness:** 2
**Presentation:** 3
**Contribution:** 2
**Rating:** 6
**Confidence:** 3

**Summary:**

This work tackles the difficulty of deploying large language models (LLMs) on resource-constrained devices by introducing a “phase-aware” quantization method that adapts precision to different inference stages, unlike traditional uniform quantization approaches. The authors develop Progressive Mixed-Precision Decoding (PMPD), a technique that lowers precision progressively as the generated sequence deepens, using schedulers to adapt precision based on tasks or prompts. Experimental results demonstrate that PMPD provides substantial speedups on Nvidia GPUs and optimized NPUs, while preserving output quality and surpassing the performance of both fp16 and uniform quantization models.

**Strengths:**

- The author discusses many design choices, with their pros and cons.
- The discussion on the tolerance of quantization in different depths of the decoding stage is informative

**Weaknesses:**

- The paper does not discuss much about how they do the precision switching in their implementation, which could be a potential risk of memory usage (loading the same model with different precision) or additional latency (loading the model weight on the fly).
- The accuracy results are all based on MHA models, lacking discussion on other types of models like GQA models.

**Questions:**

1. How do you implement the precision switch operation during the decoding stage? And what is the actual cost of such switching?

2. How's the result on Llama3? Or other GQA models?

3. How's your method compared with static quantization methods like GPTQ and AWQ both for accuracy and efficiency?

---

> ### Author Response · Authors · 2024-11-21
>
> Dear Reviewer A5Dp,
>
> We thank you for your time and your detailed feedback. Please check our reponses below.
>
> Q1: Implementation and cost of precision switching.
>
> As described in Sec. 2.1, PMPD leverages AnyPrecision LLM [3], which enables efficient storage of M different precision models (1 to M-bit) using only M bits, while preserving model accuracy. This approach ensures PMPD requires no additional memory overhead. Furthermore, precision switching incurs negligible latency since no extra weight loading is required.
>
> To verify this, we measured the decoding latency of Vicuna-7B under two scenarios:
> 1. Uniform 2-bit precision for 100 decoding steps
> 2. Mixed precision: 3-bit for 1 step followed by 2-bit for 99 steps
>
> The results, shown in the table below, confirm that the overhead from precision switching is negligible.
>
> |  | Latency (s/token) |
> | :---- | ----- |
> | Uniform 2-bit precision (no switching) | 0.0240 |
> | 3-bit for 1 step, then switch to 2-bit (precision switching) | 0.0241 |
>
> We thank the reviewer for coming up with experiments to strengthen our work. We will further clarify the implementation details and cost of the precision switching in the revised paper.
>
> Q2: Result on GQA models.
>
> As suggested by the reviewer, we performed further experiments on Llama3 that leverages GQA attentions, using static schedulers. The results confirm PMPD's ability to effectively reduce bitwidth while preserving model performance, validating its compatibility with different attention mechanisms.
>
> | Model | Method | CNN/DM  |  | DSUM |  | IWSLT |  |
> | :---- | :---- | :---- | :---- | :---- | :---- | :---- | :---- |
> |  |  | bit | Rouge-L/BERTScore | bit | Rouge-L/BERTScore | bit | BLEU/SacreBLEU |
> | Llama3-8B | 2bit | 2  | 4.35 / 75.8 | 2 | 5.58 / 81.7 | 2 | 0 / 0.166 |
> |  | 3bit | 3  | 18.3 / 85.2 | 3 | 19.0 / 85.8 | 3 | 23.1 / 23.2 |
> |  | PMPD  | 2.66 | 17.8 / 84.8 | 2.31 | 18.7 / 85.8  | 2.71 | 22.1/22.1 |
>
> Q3: Comparisons with more baselines
>
> PMPD introduces a precision allocation scheme that is independent of the underlying post-training quantization (PTQ) method. While we primarily utilize SqueezeLLM as our PTQ method, PMPD is compatible with other approaches like GPTQ.
>
> Our choice of SqueezeLLM with uniform bitwidth as the baseline is methodologically sound, as it isolates the impact of our precision allocation strategy. Direct comparisons with other PTQ methods (such as GPTQ and AWQ) would primarily reflect differences in the underlying quantization approaches—a comparison already reported in the SqueezeLLM paper [2], which demonstrated SqueezeLLM's superior performance compared to GPTQ [4] and AWQ [5].
>
> To showcase the generalizability of our method, we have already included results with GPTQ in Appendix A.4, Table 6 of our original submission.
>
> To demonstrate PMPD's versatility, we further include a comparison below showing that GPTQ underperforms both SqueezeLLM and PMPD.
>
> |  |  | CNN/DM |  | DSUM  |  | IWSLT  |  |
> | :---- | :---- | :---- | :---- | :---- | :---- | :---- | :---- |
> |  |  | bit  | (Rouge-L/BERTScore) | bit | (Rouge-L/BERTScore) | bit | (BLEU/SacreBLEU) |
> | Vicuna-7B | GPTQ (3bit) | 3 | 11.7 / 79.6 | 3 | 6.96 / 81.7 | 3 | 1.5 / 1.5 |
> |  | SqueezeLLM (3bit) | 3 | 24.2 / 86.9 | 3 | 24.4 / 88.2  | 3 | 31.6 / 31.6 |
> |  | PMPD using SqueezeLLM  | 2.39 | 24.3 / 87.0 | 2 | 25.0 / 88.2  | 2.37 | 31.0 / 31.1 |
>
> For efficiency, since the underlying PTQ method we used in Experiment is SqueezeLLM, here is the comparison with GPTQ based on results from SqueezeLLM[2].
>
> | bit  |  | Speedup  |
> | :---- | :---- | :---- |
> | 4 | GPTQ (3bit) | 2.3 |
> |  | SqueezeLLM (3bit) | 2.1 |
> | 3 | GPTQ (3bit) | 2.0 |
> |  | SqueezeLLM (3bit) | 1.8 |
>
> [2] SqueezeLLM: Dense-and-Sparse Quantization. ICML 2024.
>
> [3] Any-Precision LLM: Low-Cost Deployment of Multiple, Different-Sized LLMs. ICML 2024
>
> [4] GPTQ: Accurate Post-Training Quantization for Generative Pre-trained Transformers
>
> [5] AWQ: Activation-aware Weight Quantization for LLM Compression and Acceleration

---

> > ### Comment · Area_Chair_eGvj · 2024-11-24
> > **discussion**
> >
> > Dear Reviewer A5Dp -- could you please look at the authors' rebuttal and acknowledge that you've read it?  Also, if you have any further questions for the authors, please let them know ASAP.

---

### Official Review · Reviewer_RDtY · 2024-11-04

**Soundness:** 3
**Presentation:** 2
**Contribution:** 2
**Rating:** 6
**Confidence:** 4

**Summary:**

The authors point out that different stages of LLM inference require different quantization bit-widths. In oracle experiments, they discovered that increasing the model weight bit-width during the prefill phase can enhance model performance. They also find that using a higher bit-width in the early stages of decoding helps maintain model performance. Based on these findings, the authors search for different quantization bit-widths for the prefill and decode stages. Additionally, they use an optimized scheduler to gradually reduce the model bit-width during decode stage. Using these two schemes, the quantized model can ensure output quality on the challenging MT-Bench task. Meanwhile, the quantized model achieves a throughput improvement of 3.8-8.0× on NPU and a linear layer acceleration of 1.4-12.2× on GPU.

**Strengths:**

1. Compared to previous methods of mixed-precision quantization within models, this paper innovatively considers introducing mixed-precision quantization at different stages of LLM inference.

2. This paper not only validates the algorithmic performance of the proposed method but also measures the hardware performance like speedup ratio and throughput on both GPUs and NPUs.

**Weaknesses:**

1. In Section 3.1 of this paper, the experiment results are insufficient to demonstrate the validity of Motivation One.

    - First, the authors only increase the bit-width during the prefill stage and found that model performance was better than that when using uniform bit-width, but do not increase the bit-width during the decode stage to evaluate model performance. Therefore, the results do not demonstrate that the prefill and decode stages have different error resilience.

    - Secondly, this section only presents the model's outputs for three specific inputs, lacking quantitative results.

    - It is recommended that the authors supplement the results of high-precision decoding with low-precision prefill and provide performance of the model on MT-Bench to support Motivation One.

2. In Section 4.1 of this paper, the authors search for the optimal bit-widths for the prefill and decode stages assuming that the decode stage is conducted with a uniform bit-width. However, during actual inference, the model bit-width in the decode stage gradually decreases. Therefore, it cannot be guaranteed that the previously found results remain optimal. Then what is the significance of this search step? Is it possible to directly use the bit-width of the prefill stage as the starting point for the bit-width decrease in the decode stage?

3. In Section 4.3 of this paper, the authors design the Task-Agnostic Learned Scheduler to predict the steps at which the bit-width should decrease during the decode stage. Theoretically, the optimal precision switch points may depend on both the model's inputs and the outputs. However, this scheduler only uses the token features generated during the prefill stage as input. Is it reasonable to overlook the information from the model's generated content?

4. The experimental results in this paper are not comprehensive enough. First, the maximum parameter size of the model used in the experiments is 7B, and larger models are not evaluated. Secondly, the paper only provides GPU speedup ratios for the linear layers and does not present end-to-end speedup data.

5. This paper has misunderstandings regarding some concepts. For example, the maximum context length of LLMs (CL) refers to the sum of the model's maximum input and output lengths, but in Section 4.3, the paper incorrectly asserts that CL is the model's maximum output length.

**Questions:**

See weaknesses

---

> ### Author Response · Authors · 2024-11-21
>
> Dear Reviewer RDtY,
>
> We thank you for your time and your detailed feedback. Please check our reponses below.
>
> Q1: quantitative analysis about the optimal bitwidth allocation between prefilling and decoding stages.
>
> We performed experiments comparing different bitwidth configurations for prefilling and decoding stages using MT-Bench. The fp16 model outputs were used as the reference answer to compute the Rouge-L score:
>
> | Decoding/Prefill Strategy | Rouge-L |
> | :---- | :---- |
> | 2bit Prefill, 2bit Decode | 18.3 |
> | 3bit Prefill, 2bit Decode | 28.1 |
> | 2bit Prefill, 3bit Decode | 22.2 |
>
> These findings complement our quantitative analysis presented in Fig. 6 (Section 5.3), which we will further elaborate in the revised paper.
>
> The results demonstrate that, given the same total bitwidth budget, allocating higher bitwidth to prefilling (3-bit prefill, 2-bit decode) significantly outperforms the alternative configuration (2-bit prefill, 3-bit decode). This validates our proposed approach. Additionally, as discussed in our paper, our high-precision-prefilling strategy is also more hardware-efficient since prefilling operations are compute-bound [1].
>
> [1] LLM Inference Unveiled: Survey and Roofline Model Insights. Yuan et al. 2024
>
> Q2: Bitwidth search for prefilling did not consider that the decoding precision was decreasing.
>
> Our research demonstrated that higher precision during prefilling enhanced both performance and hardware efficiency. Given this observation, there may be different approaches to prefilling bitwidth search. In our paper, we chose a sequential optimization solution and used the prefill stage bitwidth as the initial point for decoding as it is simple yet empirically effective. However, we acknowledge that more advanced methods could be explored. The reviewer suggests an expanded search space that could incorporate non-uniform bitwidths in the decoding stage, an interesting direction for future investigation. We appreciate this insightful suggestion and will explore it in the future research.
>
> Q3: The Task-Agnostic Learned Scheduler did not make use of the generated tokens.
>
> We agree with the reviewer that ideally we should also consider the generated tokens for optimal switch point search. We currently only used the input prompt tokens for two practical reasons: i) predicting a precision at every decoding step introduces extra latency, which may offset the benefits of model quantization. In contrast, our input-based approach maintains constant scheduler overhead. ii) our empirical results showed that input prompt information alone yielded effective results.
>
> That being said, we acknowledge that incorporating generated content into scheduler decisions is a promising future direction that we will explore in the future research.

---

> > ### Comment · Reviewer_RDtY · 2024-11-23
> > **Thanks for the rebuttal**
> >
> > Thank you for the detailed feedback and additional experiments. The authors have addressed most of my concerns, so I will increase my score to reflect this.
> >
> > Additionally, I have one more suggestion that might help make this paper more solid:
> >
> > **A deeper analysis of MT-Bench:** My original intention was to highlight that MT-Bench serves as a benchmark for evaluating the open-ended generation of LLMs, which is closer to the topic of this paper. The MT-bench uses a strong LLM judger to evaluate the quality of the responses generated by LLMs, which may better reflect the semantic quality of the generated text (something that metrics like ROUGE-L may struggle to achieve). Furthermore, given that MT-Bench contains only 80 samples, including a small-scale human evaluation could further enhance the reliability of the experiments.
> >
> > Additionally, I would suggest a more detailed analysis of the outputs from quantized LLMs in the appendix. For instance, after quantization, LLMs might exhibit some errors like failure to generate outputs, token repetition, logic errors, and so on. The authors could manually categorize and analyze these issues, providing specific examples to illustrate how the proposed method improves performance in the open-ended generation benchmark.

---

> > > ### Author Response · Authors · 2024-11-25
> > >
> > > Dear Reviewer RDtY,
> > >
> > > Thank you for your prompt response and insightful suggestions to improve our paper.
> > >
> > > Below, we present the LLM-Judge scores evaluated using OpenAI’s GPT-4. PMPD demonstrates its ability to reduce bitwidth redundancy while preserving overall performance:
> > >
> > >
> > > | Model                 | Bitwidth | Writing | Roleplay | Reasoning | Math  | Coding | Extraction | STEM   | Humanities | Overall |
> > > |------------------------|----------|---------|----------|-----------|-------|--------|------------|--------|------------|------------|
> > > | MobileLLaMA (PMPD)   | 3.68  | 3.10    | 3.80     | 1.45      | 1.00  | 1.45   | 1.05       | 3.50   | 4.85       |2.525 |
> > > | MobileLLaMA (Baseline)   | 4.0 | 3.05    | 4.15     | 1.45      | 1.00  | 1.25   | 1.10       | 2.80   | 4.20       |2.375 |
> > >
> > >
> > > Due to constraints related to obtaining approval for human subject evaluations and limited time, we leave a comprehensive human evaluation to future work. Nevertheless, we appreciate the reviewer’s suggestion and have included examples showcasing both a successful and a failed case from the PMPD model, along with detailed commentary.
> > >
> > > Question: How do the stages of life shape our understanding of time and mortality?
> > >
> > > Answer: The stages of life can shape our understanding of time and mortality in several ways.
> > >
> > > 1. Childhood: Children are often more focused on the present moment and less concerned with the future. This can lead to a more relaxed and flexible understanding of time and mortality.
> > > 2. Adulthood: As we enter into adulthood, we become more aware of the passage of time and the possibility of death. This can lead to a more pessimistic view of time and mortality.
> > > 3. Midlife: Midlife is a time of significant life changes, including career changes, family responsibilities, and the possibility of death. This can lead to a more realistic and optimistic view of time and mortality.
> > > 4. Later life: As we age, we may experience changes in our lives and the possibility of death. This can lead to a more realistic and pessimistic view of time and mortality.
> > >
> > > Overall, the stages of life can shape our understanding of time and mortality by shaping our understanding of the present moment and the possibility of death.
> > >
> > > Comment: The PMPD model is able to produce sensible results for humanity subjects.
> > >
> > > Question: When a number is divided by 10, the remainder is 4. What is the remainder when twice the number is divided by 4?
> > >
> > > Answer: To find the remainder when a number is divided by 4, you can use the following steps:
> > >
> > > 1. Write down the number you want to find the remainder of.
> > > 2. Write down the number you want to find the remainder of again.
> > > 3. Multiply the number you wrote down 2 times.
> > > 4. Write down the result.
> > >
> > > Comment: The result is not very relevant to the question.
> > >
> > > We will consider including additional analyses of outputs from quantized LLMs in the appendix if it provides further insights into our approach.
> > >
> > > Thank you again for your time.
> > >
> > > Best regards,
> > >
> > > Anonymous Authors

---

> > > > ### Comment · Reviewer_RDtY · 2024-11-29
> > > >
> > > > Thank you for sharing the additional results and nice examples. I have no further concerns and will maintain my positive score.

---

> ### Author Response · Authors · 2024-11-21
>
> Q4.1: Results for models larger than 7B.
>
> As mentioned in our Introduction (Sec. 1), PMPD mainly targets edge and mobile applications. We agree with the reviewer that it would be interesting to see if larger models could also benefit from PMPD. We conducted additional experiments using Llama3-8B and longchat-16k-13B models, with a static scheduler for PMPD given the availability of validation data. These new results demonstrate our method's generalizability across model scales.
>
> | Model | Method | CNN/DM  |  | DSUM |  | IWSLT |  |
> | :---- | :---- | :---- | :---- | :---- | :---- | :---- | :---- |
> |  |  | bit | Rouge-L/BERTScore | bit | Rouge-L/BERTScore | bit | BLEU/SacreBLEU |
> | Llama3-8B | 2bit | 2  | 4.35 / 75.8 | 2 | 5.58 / 81.7 | 2 | 0 / 0.166 |
> |  | 3bit | 3  | 18.3 / 85.2 | 3 | 19.0 / 85.8 | 3 | 23.1 / 23.2 |
> |  | PMPD  | 2.66 | 17.8 / 84.8 | 2.31 | 18.7 / 85.8  | 2.71 | 22.1 / 22.1 |
> | longchat-13B-16k | 2bit | 2 | 9.29 / 72.8 | 2 | 10.6 / 80.8 | 2 | 2.94 / 2.94 |
> |  | 3bit | 3 | 19.9 / 86.4 | 3 | 19.9 / 86.6 | 3 | 21.6 / 21.6 |
> |  | PMPD | 2.37 | 19.9 / 86.4 | 2 | 20.3 / 86.5 | 2.71 | 21.4 / 21.4 |
>
> Q4.2: End-to-end GPU speedup data.
>
> As PMPD built on SqueezeLLM [2] and Any-precision LLM [3], we evaluated PMPD's GPU performance primarily on linear layers, following common practices in the literature [2][3]. To measure end-to-end latency improvements compared to fp16 models, we tested using single-batch prompts with a context length of 1024 and recorded Self-CUDA-time as in our paper. Our results demonstrate that PMPD consistently outperforms uniform quantization baselines across various models and GPU architectures.
>
> | GPU | Method | Vicuna-7B | MobileLlama | Phi-1.5 | Zephy-3B |
> | :---- | :---- | :---- | :---- | :---- | :---- |
> | 4090 | Baseline-h | 3.05 $\\times$ | 1.80 $\\times$ | 1.75 $\\times$ | 2.07 $\\times$ |
> |  | PMPD | 3.36 $\\times$ | 1.90 $\\times$ | 1.83 $\\times$ | 2.22 $\\times$ |
> | A40 | Baseline-h | 2.67 $\\times$ | 1.71  $\\times$ | 1.66 $\\times$ | 1.91 $\\times$ |
> |  | PMPD | 2.91 $\\times$ | 1.82 $\\times$ | 1.74 $\\times$ | 2.07 $\\times$ |
>
> While the overall speedup is more modest than when evaluating linear layers alone, it is expected since both PMPD and the baselines share common fp16 operations, such as self-attention, which are often time-consuming. Notably, NPUs present a more promising platform than GPUs for maximizing PMPD's performance benefits on edge devices, as they better support custom algorithmic operations.
>
> [2] SqueezeLLM: Dense-and-Sparse Quantization. ICML 2024.
>
> [3] Any-Precision LLM: Low-Cost Deployment of Multiple, Different-Sized LLMs. ICML 2024
>
> Q5: The vague usage of maximum context length of LLMs (CL)
>
> We thank the reviewer for pointing out the vague use of the term. We will revise the paper as suggested.

---

### Author Response · Authors · 2024-11-21

We thank the reviewers for reading our paper and providing valuable feedback. We are encouraged that the strengths of our paper were commonly acknowledged by the reviewers: the innovative approach (Reviewers RDtY, TCS8, FUCY), comprehensive experiments demonstrating both output quality and hardware efficiency (Reviewers RDtY, TCS8, FUCY), and informative discussions and ablation studies (Reviewers A5Dp, TCS8). We answer each reviewer's comments and questions in detail below, and will incorporate their suggestions in the revised paper.

---

### Meta-Review · Area_Chair_eGvj · 2024-12-21

**Metareview:**

This paper introduces a phase-aware quantization strategy for LLM inference, allocating higher precision to the prefill stage and adaptively reducing precision during decoding. Its core strengths lie in demonstrating consistent throughput gains. Initially, concerns were raised regarding writing, limited benchmarks, insufficient details, uncertainty about choosing the hyperparameter N, etc. However, the authors’ rebuttal effectively resolved these points, offering additional experiments, clarifications, and corrections—thereby strengthening the overall contribution.

**Additional Comments On Reviewer Discussion:**

see above

---

### Decision · Program_Chairs · 2025-01-22

Accept (Poster)